# *Arabidopsis* Ubiquitin-Conjugating Enzymes UBC7, UBC13, and UBC14 Are Required in Plant Responses to Multiple Stress Conditions

**DOI:** 10.3390/plants9060723

**Published:** 2020-06-08

**Authors:** Hui Feng, Sheng Wang, Dengfeng Dong, Ruiyang Zhou, Hong Wang

**Affiliations:** 1College of Agriculture, Guangxi University, Nanning 530005, China; huf524@mail.usask.ca (H.F.); dengfengdong6@gmail.com (D.D.); 2Department of Biochemistry, Microbiology and Immunology, University of Saskatchewan, Saskatoon, SK S7N 5E5, Canada; sheng.wang@usask.ca

**Keywords:** *Arabidopsis*, ubiquitin-conjugating, UBC7, oxidative stress, salt stress, stress response

## Abstract

Protein ubiquitination plays important roles in plants, including stress responses. The ubiquitin (Ub) E2 enzymes are required in the transfer of Ub to a substrate and are also important in determining the Ub-chain linkage specificity. However, for many of the 37 E2 genes in *Arabidopsis thaliana*, there is currently little or no understanding of their functions. In this study, we investigated three members of an E2 subfamily. The single, double, and triple mutants of *UBC7*, *UBC13*, and *UBC14* did not show any phenotypic changes under normal conditions, but were more sensitive than the wild-type (WT) plants to multiple stress conditions, suggesting that the three genes are not critical for normal growth, but required in plant stress responses. The severity of the phenotypes increased from single to triple mutants, suggesting that the functions of the three genes are not completely redundant. The three E2s are closely related to the yeast Ubc7 and its homologs in animals and human, which are an important component of the endoplasmic reticulum (ER)-associated degradation (ERAD) pathway. The stress sensitivity phenotypes of the mutants and shared evolutionary root with the Ubc7 homologs in yeast and metazoans suggest that UBC7, UBC13, and UBC14 may function in the plant ERAD pathway.

## 1. Introduction

The control of the level of proteins (especially regulatory proteins) according to the needs of cellular processes and in the response to various stresses is essential for all organisms. Protein ubiquitination is a fundamental mechanism of post-translational modifications with vast functional roles in eukaryotes including plants, and is responsible for most of the regulated protein degradation through the ubiquitin–proteasome system (UPS) [1,2,3]. In protein ubiquitination, a substrate protein is modified with the attachment of one ubiquitin (Ub) (monoubiquitination) or one poly-Ub chain (polyubiquitination). The attachment of a Ub or Ub chain requires a cascade of three Ub-associated enzymes, namely the E1 enzyme for Ub activation, the E2 enzyme for Ub-conjugating, and the E3 enzyme for transferring Ub from an E2 to substrate proteins [4]. The E2s are required in the transfer or conjugation of the Ub to a substrate and important in determining whether the Ub modification of a substrate is monoubiquitination or polyubiquitination [5]. Some E2s directly mediate the transfer of the Ub to the substrate protein, whereas other E2s transfer the Ub to the substrate protein through an E3 ligase. Furthermore, different Ub modifications or chains could confer different fates to a substrate.

The *Arabidopsis* genome has at least 37 genes encoding ubiquitin E2s [1,6], compared to 10 in budding yeast [7] and 30 in human [5]. The greater number of E2s in *Arabidopsis* may reflect the needs for greater functional diversity. Almost all *Arabidopsis* E2s (35 out of 37) have been shown to possess the classical and generic E2 activity in forming a thioester-linkage between Ub and the E2 [6,8]. Additionally, some understanding has been gained on the functions of certain E2s in plants and their biochemical properties [1]. However, for many of the E2s in plants, there is little or no information regarding their functions. Considering the fundamental roles of protein ubiquitination, it is important to understand the functions of those E2s.

Research so far has identified important roles for certain E3 ligases and other components of the UPS in plant stress responses [9,10,11]. Protein ubiquitination can promote the degradation of a substrate protein. However, the consequence to the plant response to a particular stress or environmental change depends on whether the substrate protein is a positive or negative regulator of the process [9]. Therefore, the specific effect of protein ubiquitination largely depends on the functions of substrate proteins and the cellular processes they are involved in. Since the E2s are required in the transfer of Ub to a substrate and are also important in determining the length and linkage specificity of a Ub chain and reside selection on a substrate [5], it is conceivable that plant stress responses would require the involvement of at least some E2s. Experimental evidence has implicated some plant E2 genes in the stress responses [12]. It has been shown that many of the genes encoding the putative E2 enzymes in rice and potato genomes are induced by stresses such as salt and cold as well as by abscisic acid (ABA) [13,14].

Among the *Arabidopsis* E2s, UBC32, UBC33, and UBC34 (three members of subfamily XIV) and a few others have been found to be involved in plant responses to abiotic stresses. The *ubc32* knockout mutant had increased tolerance to oxidative and salt stresses [15,16]. Furthermore, the mutants and particularly the triple mutant of *UBC32*, *UBC33*, and *UBC34* had increased drought tolerance compared to the wild-type (WT) plants [17]. Several lines of evidence indicate that UBC32 is a component of the plant endoplasmic reticulum (ER)-associated degradation (ERAD) pathway and has some functional resemblance to yeast Ubc6 in addition to the sequence similarity [16,18]. ERAD is a conserved protein quality control mechanism in eukaryotes to remove misfolded proteins in the ER and maintain protein homeostasis [19,20]. It is worth noting that the yeast Ubc6 is a main component of the ERAD pathway and functions mainly with the E3 ligase Doa10 to ubiquitinate and remove ERAD substrates. Regarding other *Arabidopsis* E2s, the mutants of *UBC26* showed increased sensitivity to ABA and the results suggest that UBC26 along with RFA4 E3 ligase regulate the level of ABA receptors [21]. In addition, *UBC35* and *UBC36* (also referred to as *UBC13A* and *UBC13B*) were found to be involved in the response to pathogen infection and low temperature stress by regulating programmed cell death [22]. Despite the noted progress, for the majority of the E2s in *Arabidopsis*, their involvement in plant stress responses remains unknown.

*Arabidopsis UBC7*, *UBC13*, and *UBC14* are three members in the subfamily V of the 37 E2 genes [1]. It was demonstrated years ago that these E2 proteins could catalyze poly-Ub chain formation in vitro, but plants overexpressing *UBC7* had no phenotype [23]. There has been no report on their functions in plants. We present several lines of evidence to show that the three E2s are important in plant responses to multiple stress conditions.

## 2. Results

### 2.1. Characterizing UBC7, UBC13, and UBC14 Single Mutants, and Creating Double and Triple Mutants

Genomic-wide gene expression data show that *UBC7, UBC13*, and *UBC14* are expressed widely in different tissues of *Arabidopsis thaliana*, and their expression levels are similar in most tissues, except for the inflorescence and seeds in which *UBC14* has a higher level than the other two members of the subfamily (Appendix A). To understand their functions, we obtained individual T-DNA mutants for *UBC7* (*At5g59300*), *UBC13* (*At3g46460*), and *UBC14* (*At3g55380*) and then homozygous progeny lines (Figure 1A). PCR using genomic DNA with a pair of gene-specific primers amplified a DNA band of the expected size for each of the genes from the WT control, but not from the T-DNA mutants (Figure 1B), while a mutant DNA band was amplified with a primer specific to the left border of the T-DNA and a primer specific to the gene. We also used reverse transcription PCR (RT-PCR) to determine gene expression. The results showed that a transcript band of expected size was amplified using a pair of gene-specific primers from the WT, but not from the mutant plants of the respective genes (Figure 1C), indicating that each respective gene was disrupted in these mutants. These mutants are thus referred to as *ubc7-1* (for SALK_111680), *ubc13-1* (GK-866B08 and ABRC seed stock number CS483060), and *ubc14-1* (for SALK_119217).

In this study, *UBC13* refers to *At3g46460* as it was named originally [23]. It needs to be noted that previously *UBC13* was also given to two subtype XV E2 genes, named *UBC13A* (*At1g78870*) and *UBC13B* (*At1g16890*) (*UBC35* and *UBC36*), following the nomenclature of yeast and animal homologs based on the function and sequence relativeness to Ubc13 genes in other species [24].

Since *UBC7*, *UBC13*, and *UBC14* are three closely related members [1,23], there is possible functional redundancy among them. Therefore, from crossing the single mutants, we created three different double mutants (*ubc7-1 ubc13-1*, *ubc7-1 ubc14-1*, and *ubc13-1 ubc14-1*), and for simplicity, they are referred to as *ubc7/13*, *ubc7/14*, and *ubc13/14*. Furthermore, using the double mutants, we created the triple mutant *ubc7/13/14*. For each mutant, we obtained multiple lines derived from the initial crosses.

### 2.2. UBC7, UBC13, and UBC14 Mutants Showed Increased Sensitivity to Salt, Oxidative Stress, and ABA

When grown in soil under normal conditions, all of the single, double, and triple mutants were indistinguishable from the WT plants in terms of plant size, leaf and flower morphology, and seed setting (Appendix A). To identify a possible phenotype, we grew the triple mutant seedlings under various hormonal and stress conditions with WT seedlings grown in the same plate as the control. The mutant seedlings responded similarly to some treatments (Appendix A). For instance, on the plates containing an auxin (1-naphthaleneacetic acid (NAA)), cytokinin (kinetin), or salicylic acid (SA), the growth of the mutant and WT seedlings was similarly affected. However, the *ubc7/13/14* mutant seedlings were more sensitive to NaCl and PBI425, an analog ABA (Appendix A). The ABA analog PBI425 was used since it has been well documented that the analog functions similarly to ABA, but it is more persistent in plant cells and has stronger hormonal activity [25,26].

To quantitatively determine the difference between the mutant and WT plants, root growth was used. Additionally, since phenotypic changes of the mutant plants may depend on the level of stress, we used a range of concentrations for a particular test reagent. In each plate, WT and mutant seedlings were grown side by side. The length of primary roots was measured with ImageJ and average root length of the WT seedlings was used to normalize the root length of the mutant seedlings in the same plate, thus minimizing the variation among plates. Compared to the WT, the triple mutant seedlings had much reduced root growth when treated with NaCl, PBI425, and paraquat (Figure 2), indicating that the triple mutant was more sensitive to salt and oxidative stresses than the WT. Quantitative data showed that root growth of the mutant seedlings was more inhibited than the WT at various concentrations of NaCl (starting from 60 mM), ABA analog PBI425 (starting from 0.1 μM), and paraquat (starting from 0.015 μM) (Figure 3). Since the sensitivity to NaCl could be due to osmotic or ionic stress or both, the effect of osmotic stress using different concentrations of mannitol was also examined. The mutant seedlings were only slightly more sensitive to 1.0% and 2.0% of mannitol (Figure 2 and Figure 3). These results showed that the triple mutant plants had much stronger sensitivities than the WT to salt, oxidative stress, and ABA (analog) with a slightly increased sensitivity to mannitol (osmotic stress).

To determine functional redundancy of the three genes, single, double, and triple mutants were analyzed on control ½ MS agar plates or plates containing 150 mM NaCl. On the control plates, all mutant seedlings grew similarly to the WT seedlings (Appendix A). On the 150 mM NaCl plates, the cotyledons and leaves of some seedlings appeared white (bleached) after two weeks from seed plating, indicating that they were dead or dying. Interestingly, all three single mutants were more sensitive than the WT, and the sensitivity increased from single to triple mutants, with the triple mutant seedlings being the most sensitive (Figure 4), indicating that inactivating each gene has an additive effect on the sensitivity to salt stress. In addition, there was a slightly higher percentage of bleached seedlings for the *ubc7* single mutant than for the *ubc13* and *ubc14* single mutants, suggesting that *UBC7* may have a stronger role than the other two genes.

To verify that the mutant phenotypes were due to the inactivation of the genes, a genomic *UBC7* fragment (including the putative promoter, coding, and 3′ untranslated regions) was introduced into the triple *ubc7/13/14* mutant. Genomic DNA PCR and RT-PCR results demonstrated the presence of the genomic fragment and expression of *UBC7* in four independent complementation lines (Appendix A). When the seedlings of the complementation T2 lines were subjected to salt stress, they showed increased tolerance compared to the triple mutant although they were still more sensitive than the WT (Figure 5), indicating that *UBC7* expression could partially rescue the triple mutant phenotype.

UBC7, UBC13, and UBC14 are three members in the subfamily V of *Arabidopsis* E2s [1]. Phylogenic analysis showed that their protein sequences are more related to the yeast Ubc7 and the homologs in animals and human than to other *Arabidopsis* E2s (Figure 6). Ubc7 and Ubc6 are two E2 enzymes required in the ERAD pathway [19]. In *Arabidopsis*, UBC32, UBC33, and UBC34 belong to a subclade of E2s closely related to the yeast Ubc6 and homologs in metazoans (Figure 6).

We also determined the sensitivity of the triple mutant to tunicamycin, a known ER stress inducer and inhibitor of protein glycosylation. The seedlings of the *ubc7/13/14* mutant clearly showed an increased sensitivity in terms of root growth compared to the WT at different concentrations of tunicamycin (from 25 to 100 ng/mL) (Figure 7A,B). The mutant also had an increased percentage of severely inhibited (dark brown and small) seedlings after two weeks of growth on the horizontal plates containing tunicamycin (Figure 7C and Appendix A).

## 3. Discussion

### 3.1. UBC7, UBC13, and UBC14 Are Important for Plant Stress Responses and Their Functions Are Not Completely Redundant

Among the 37 E2s in *Arabidopsis*, the functions for many have yet to be identified. In this study, we showed that UBC7, UBC13, and UBC14 (three members of subfamily V) are important for plant stress responses. Interestingly, none of the single, double, and triple mutants showed any obvious phenotypic changes in plant growth, size, morphology, and seed setting compared to the WT plants under normal conditions. On the other hand, they were more sensitive to multiple stress conditions, suggesting that their functions are not critical for normal plant growth, but required in plant responses to multiple stresses.

It is often encountered that when multiple members of a gene family (or subfamily) share functional redundancy, inactivation of single genes is not sufficient to bring about phenotypic changes. All single, double, and triple mutants of *UBC7*, *UBC13*, and *UBC14* were more sensitive to stress conditions such as salt and oxidative stress as well as to ABA. Furthermore, the sensitivity to the stress treatments increased from the single to double and triple mutants, suggesting that they share functional redundancy and that, at the same time, their functions do not completely overlap with each other. If their functions are completely redundant, the single *ubc7, ubc13*, and *ubc14* mutants would not be expected to show clear phenotypes as observed. Since the three genes were likely originated by gene duplication relatively recently in the evolutionary timeline as suggested by the phylogenetic analysis (Figure 6), it is unlikely that the three E2s had developed major differences in their fundamental biochemical properties such as the interacting E3 and associated proteins. It is more likely that the presence of three closely related genes, instead of one, could provide stronger stress response and increased capability of ERAD as well as differential regulation of their expression. In this regard, it is noted that *UBC14* appears to have a higher level of expression in the inflorescence and seeds than the other two genes (Appendix A).

### 3.2. UBC7, UBC13, and UBC14 likely Function in Plant ERAD Pathway

The 37 *Arabidopsis* E2s are grouped into 14 subfamilies [1]. Phylogenetic analysis of E2 sequences from seven representative species covering yeast, insects, mammals, and plants has shown that there are 16–18 ancestral genes in the eukaryotes, with plants and mammals sharing 16 of them while plants and yeast are sharing 12 of them [7], indicating that the homologs among different species in each of the gene family (E2 subfamily) likely share a conserved basic function. Consistently with this suggestion, many plant E2s share some functional similarities with the homologs in yeast and other species, including UBC32 similar to yeast Ubc7 in the ERAD pathway [15], UBC35/36 (UBC13A/UBC13B) to yeast and human Ubc13 in catalyzing K63-mediated ubiquitination and DNA damage response [24], and UBC22 to human Ube2S in catalyzing K11-mediated ubiquitination [27].

The shared evolutionary root and sequence similarity with yeast Ubc7 and its homologs in animals (Figure 6) suggest that UBC7, UBC13, and UBC14 may function in the plant ERAD pathway. In eukaryotes, numerous proteins are secretory proteins that are synthesized on ER-bound ribosomes, translocated into the ER lumen, and folded into the correct structures before being transported to the Golgi apparatus for secretion [28]. However, protein folding is an error-prone process particularly under stress conditions. To maintain protein homeostasis, eukaryotic cells possess several protein quality control mechanisms including ERAD in reducing misfolded proteins [19,20]. The ERAD pathway has four distinct but coupled steps for substrate recognition, substrate translocation across the lipid bilayer, polyubiquitination, and proteasome-mediated degradation [29]. Furthermore, depending on where the misfolded structure occurs, there are three separate pathways, i.e., ERAD-L, ERAD-M and ERAD-C, for recognizing and ubiquitinating substrates in the lumen, membrane, and cytosolic domains of the ER [30,31].

In yeast and mammalian cells, ERAD requires two different E2s, namely Ubc6 (human homolog Ube2J) and Ubc7 (human homolog Ube2G), which have distinct functions. In yeast, Ubc7 works with the E3 Hrd1 complex to remove misfolded ERAD-L and ERAD-M substrates [30,31], whereas both Ubc6 and Ubc7 function with Doa10 E3 ligase to remove ERAD-C substrates and, to some extent, ERAD-M substrates [32,33]. Ubc6 and Ubc7 may be responsible for catalyzing distinct reactions, as it has been suggested that Doa10 works first with Ubc6 to attach a single Ub to the substrate and then with Ubc7 to extend the Ub chain (thus polyubiquitinating the substrate) [34]. The human Ube2J1 (a Ubc6 homolog) and Ubc2G2 (a Ubc7 homolog) also have major differences in their biochemical properties and functions [35,36]. In plants, UBC32 is a homolog of the Ubc6 and is the only known ubiquitin E2 functioning in ERAD [16]. It is likely that UBC7, UBC13, and UBC14 have a function in the plant ERAD pathway similar to the yeast and mammalian homologs.

Their function in ERAD is supported by and can explain the phenotypes observed. In plants, various stresses including salinity, high temperature, drought, and pathogen infection can evoke ER stress and increase protein misfolding [37,38]. Although other modes of function for these three E3s cannot be excluded, it is conceivable that the *UBC7*, *UBC13*, and *UBC14* function in the ERAD pathway to ubiquitinate and remove certain misfolded ERAD substrates, which increase under various stress conditions. When they are inactivated, the mutant plants become more sensitive to multiple stress conditions.

Thus, this study discovered important roles of these three E2 genes in plant responses to multiple stresses, pointed to a possible fundamental mechanism, and provided important leads for further research. It will be important to investigate the molecular and cellular aspects of their functions.

## 4. Materials and Methods

### 4.1. Plant Materials

The T-DNA insertion mutants SALK_111680 (*ubc7-1*), GK-866B08 (*ubc13-1*), and SALK_119217 (*ubc14-1*) (all mutants in Col-0 background) [39,40] were obtained from the Arabidopsis Biological Resource Center (https://abrc.osu.edu/; Ohio State University, USA), and homozygous plants were identified (Figure 1). Homozygous double mutants were obtained by crossing the single mutants and genotyping the F2 populations. The triple *ubc7/13/14* mutant was obtained in the F2 population from the cross between the *ubc7/14* double mutant and the *ubc13* single mutant. From each cross, several different lines were obtained. *Arabidopsis thaliana* ecotype “Columbia-0” and mutant lines were grown in a growth room or chamber (20 °C constant, 16/8 h day/night photoperiod with a fluence rate of 90 ± 10 µmoles/m^2^/min).

### 4.2. Genotyping

Genomic DNA samples were prepared from seedlings as described [41]. Genomic PCRs were performed using the genomic DNA, gene-specific primers for the WT allele, and a gene-specific primer and a T-DNA specific primer at the left border of the T-DNA for the mutant allele, as shown in Figure 1. The primers used are listed in Appendix A. PCR reactions were typically performed for 32 cycles. A homozygous mutant plant would show only a band of the T-DNA mutant allele, not a band of the WT gene allele, for a specific gene.

### 4.3. Isolation and Analysis of Plant RNA

Total RNA samples were isolated from two-week-old young seedlings using the TRIzol reagent (formerly Invitrogen and currently ThermoFisher: https://www.thermofisher.com/ca/en/home/brands/thermo-scientific.html) following the manufacturer’s instructions. The RNA concentration was measured with a NanoVue Plus Spectrophotometer (formerly GE Life Sciences and currently Cytiva Life Sciences: https://www.cytivalifesciences.com/en/us), following the manufacturer’s instructions. cDNAs were synthesized using the ThermoScript RT-PCR system (Invitrogen). The cDNAs and gene-specific primers were used to determine the presence of the transcript of a gene by PCR. The gene *At4g33380* was used as the internal control [42].

### 4.4. Phenotyping

For observations under normal conditions, the WT and mutant plants were grown in 4-inch square pots in a growth room or chamber. They were also grown on ½ MS [43] agar plates (containing 1% sucrose and 0.7% agar). To determine the sensitivity of the mutants to various hormonal and stress conditions, NAA (auxin), kinetin (cytokinin), PBI425 (ABA analog), salicylic acid, NaCl (salt stress), paraquat (oxidative stress), and mannitol (osmotic stress) were used at the indicated concentrations. The plates were placed vertically or horizontally in a tissue culture chamber. Typically, in a vertical plate, seven WT and seven mutant seedlings were grown side by side. For a horizontal plate to analyze seedling death, 50 seeds were spot-planted on each plate. For stress phenotype analysis, four repeats were performed for each treatment. For primary root growth analysis, an image of the plate was taken on the indicated day after seed plating and the root length was measured by ImageJ (Image J 1.8.0; https://imagej.nih.gov/ij/index.html). For comparison of two means, the Student’s *t*-test was used. For comparison of multiple means, ANOVA and post-hoc Tukey test (using IBM SPSS statistics analysis 25.0, IBM, New York, USA) were performed.

### 4.5. Phylogenetics Analysis

The 37 *Arabidopsis* E2 protein sequences were collected from the TAIR (The Arabidopsis Information Resource) database (https://www.arabidopsis.org/). The sequences of rice (*Oryza sativa*) E2 homologs were obtained from the Phytozome V12 database (https://phytozome.jgi.doe.gov/pz/portal.html). The E2 sequences from baking yeast (*Saccharomyces cerevisiae*), fruit fly (*Drosophila melanogaster*), and human (*Homo sapiens*) were collected from the UniProt database (https://www.uniprot.org). The neighbor-joining phylogenetic tree was constructed by MEGA-X (version 10.1.8) [44]. Bootstrap values were calculated on 1000 iterations for testing the significance of nodes.

### 4.6. Expression Analysis using Gene Investigator

The GENEVESTIGATOR database (https://www.genevestigator.com/gv/plant.jsp) was used to examine the expression of *UBC7*, *UBC13*, and *UBC14* at different developmental stages and in different tissues [45]. The expression profiles of these genes in *Arabidopsis* are based on a large set of microarray data.

### 4.7. Construct for Mutant Complementation

For complementation, a *UBC7* genomic fragment (4000-bp), from a 1896-bp sequence upstream of ATG to a 667-bp sequence downstream of TGA, was amplified with the primers 5′-catctcgagttttcatataggatgctgc and 5′-cggtgagctccaagaaaatgaagacctcc. After the restriction digest (XhoI and SstI), the genomic fragment was cloned into a pCambia1300 vector. The construct was used to transform the *ubc7/13/14* triple mutant plants, using the infiltration method [46] with modifications [47].

## Figures and Tables

**Figure 1 plants-09-00723-f001:**
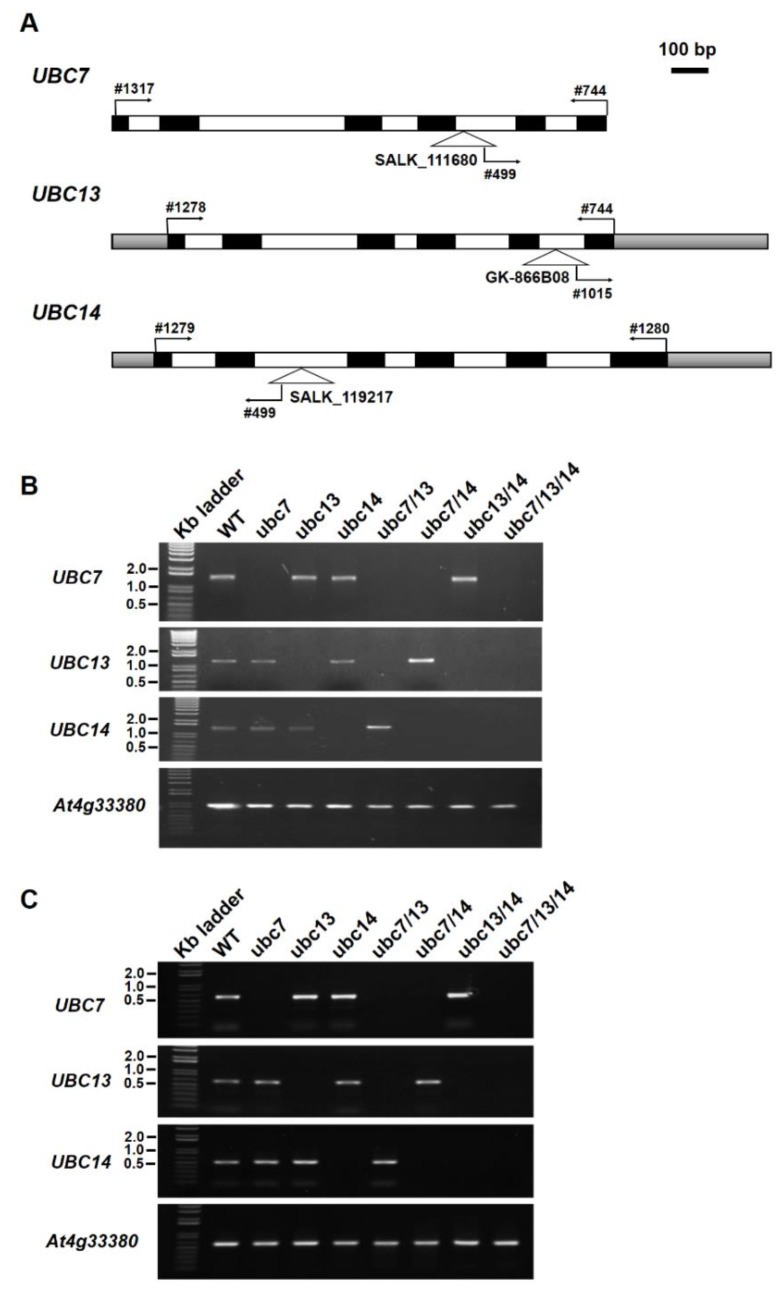
Characterization of the T-DNA mutants of *UBC7*, *UBC13*, and *UBC14*. (**A**) Schematic representation of *UBC7*, *UBC13*, and *UBC14* genomic structures with the locations of the insertion sites of T-DNA mutants shown. The gene-specific primers and a primer at the left border of the T-DNA for genotyping and RT-PCR analyses are indicated (primers are numbered). Filled boxes: exons; open boxes: introns; shadowed boxes: untranslated regions. The scale bar at the top right = 100 base pairs. (**B**) Analysis of the T-DNA lines by genomic PCR. Gene-specific primers were used to amplify the specific gene. Plant lines used are indicated above the panels. The gene *At4g33380* was used as a control to show normal amplification from the DNA samples. The sizes of a few DNA markers are indicated at the right of the panels. (**C**) Expression analysis by reverse transcription PCR (RT-PCR). Gene-specific primers were used to amplify the transcripts of each gene from the cDNA samples. Plant lines used are indicated above the panels. The gene *At4g33380* was used as a control to show normal amplification of transcripts using the cDNA samples. The sizes of a few DNA markers are indicated at the right of the panels.

**Figure 2 plants-09-00723-f002:**
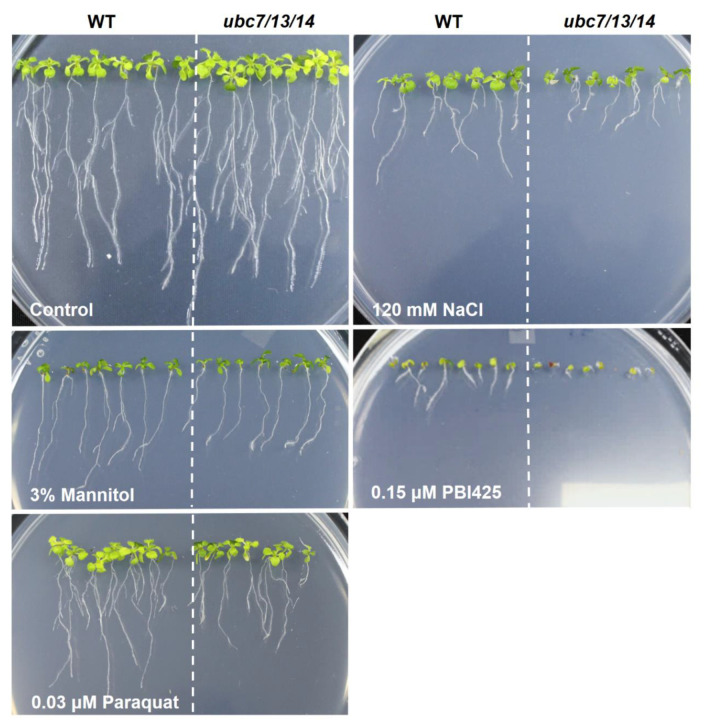
Seedling and root growth of the *ubc7/13/14* mutant seedlings under several stress conditions. The wild-type (WT) and *ubc7/13/14* mutant seeds were sterilized, stored at 4 °C, and then plated onto the normal or treatment plates with the addition of a stress-causing reagent. In each plate, seven WT and seven mutant seeds were planted side by side. The images were taken after 14 days of incubation in a plant tissue culture chamber. (**A**) Normal plate (1/2 MS + 1% sucrose + 0.7% agar), (**B**) 120 mM NaCl, (**C**) 3% mannitol, (**D**) 0.15 μM PBI425 (an abscisic acid (ABA) analog), and (**E**) 0.03 μM paraquat.

**Figure 3 plants-09-00723-f003:**
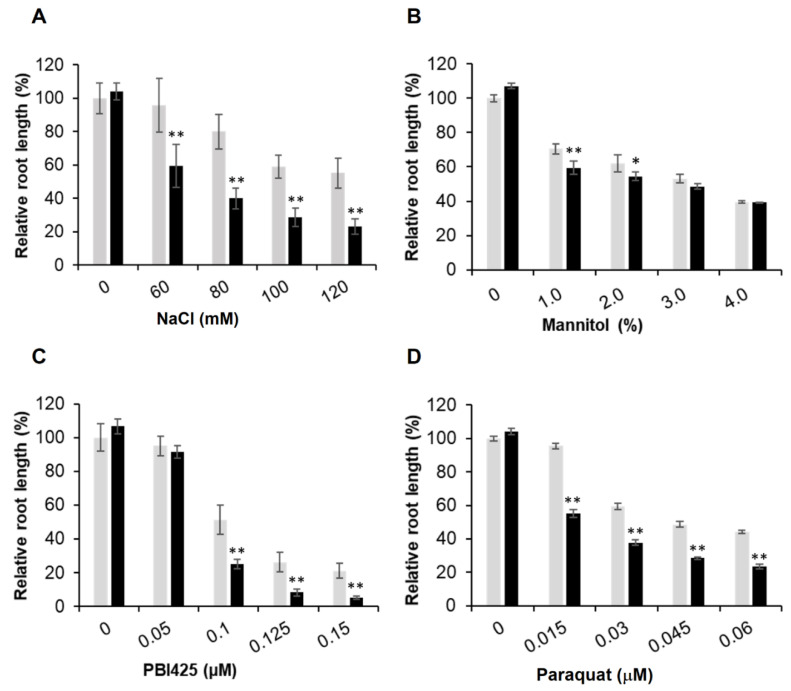
Sensitivity of the *ubc7/13/14* seedlings to different concentrations of stress reagents. The WT and *ubc7/13/14* mutant seeds were sterilized, stored at 4 °C, and then plated onto the normal or treatment plates containing the stress reagents at the indicated concentrations. In each plate, seven WT and seven mutant seeds were planted side by side. The primary root length was measured using ImageJ on the 14th day after seed plating. For easy comparisons among treatments, the root length of WT seedlings in the normal plates was set to 100%. Each treatment had five replicate plates. The significance of difference between the mutant and WT was analyzed using the Student’s *t*-test (error bar = SE; *: *p* < 0.05; **: *p* < 0.01). (**A**) 120 mM NaCl, (**B**) 3% mannitol, (**C**) 0.15 μM PBI425 (an ABA analog), and (**D**) 0.03 μM paraquat.

**Figure 4 plants-09-00723-f004:**
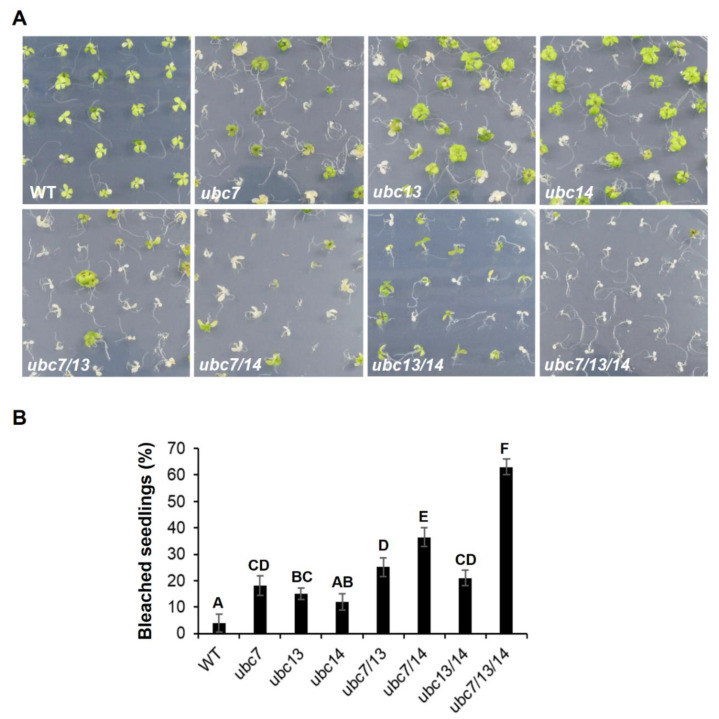
Sensitivity of single to triple *ubc7/13/14* mutants to salt stress. Seedlings of WT and mutant seedling growing on the medium plates containing 150 mM NaCl. *Arabidopsis* WT and mutant seeds were sterilized, stored at 4 °C, and then planted on the salt-containing plates (each plate having 50 seeds). The plates were placed in a tissue culture chamber. (**A**) Representative images of the plates photographed on the 14th day after seed plating. The single, double, and triple mutants are indicated as *ubc7*, *ubc13*, *ubc14*, *ubc7/13*, *ubc7/14*, *ubc13/14*, and *ubc7/13/14*. (**B**) The percentage of bleached seedlings. Each plant line had five plates with each plate having 50 seeds. Seedlings were surveyed on the 14th day after seed plating. One-way ANOVA and post-hoc Tukey test were used to analyze the data, and significant differences are indicated by different letters (uppercase at *p* < 0.01 level).

**Figure 5 plants-09-00723-f005:**
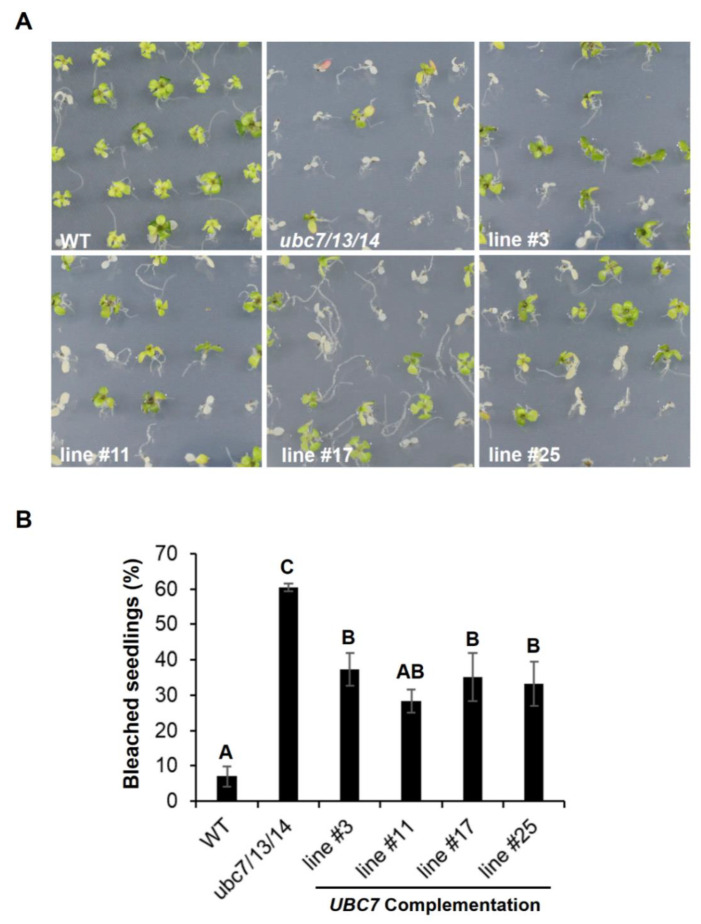
Complementation of the *ubc7/13/14* mutant by *UBC7*. A genomic *UBC7* fragment including the promoter region was introduced into the *ubc7/13/14* mutant. The seedlings of the WT control, the *ubc7/13/14* mutant, and four complementation (T2) lines were grown on medium plates containing 150 mM NaCl, and a survey of bleached seedlings was taken on the 14th day after seed plating. Each line had five plates with each having 50 seeds. One-way ANOVA and post-hoc Tukey test were used to analyze the data, and significant differences are indicated by different letters (uppercase at *p* < 0.01 level).

**Figure 6 plants-09-00723-f006:**
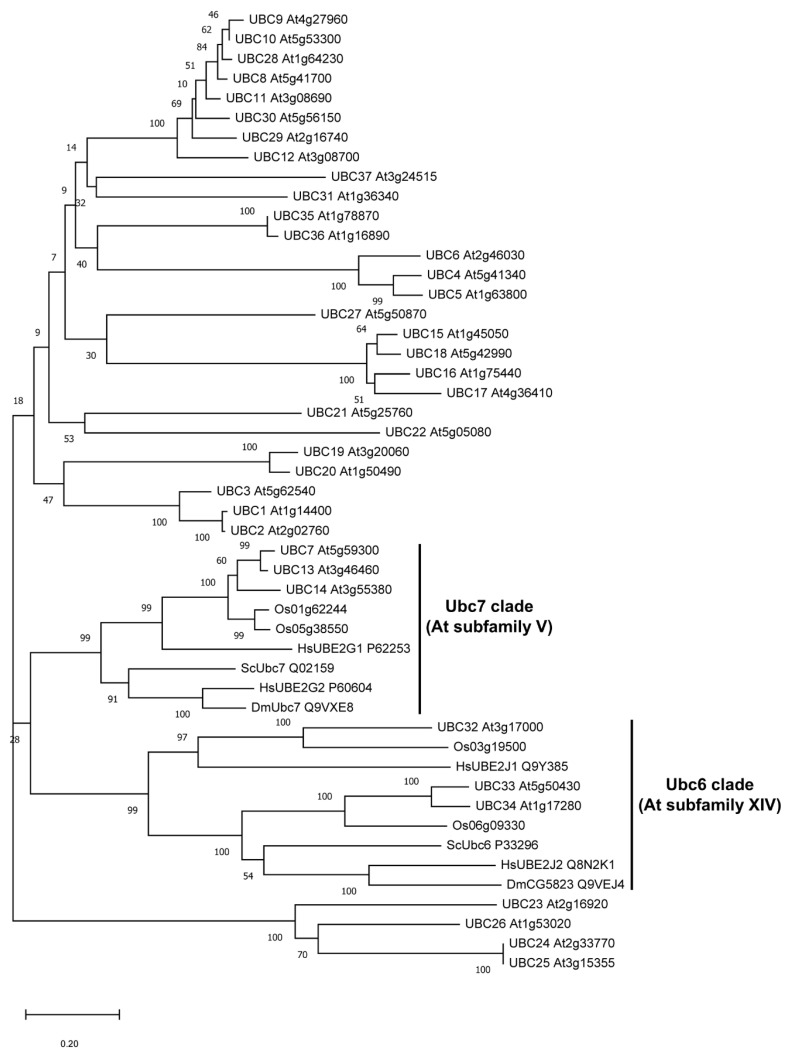
Phylogenetic analysis of *Arabidopsis* ubiquitin (Ub) E2s with sequences from several other species related to yeast Ubc6 and Ubc7. Other E2s used are from rice (*Oryza sativa*—Os), yeast (*Saccharomyces cerevisiae*—Sc), fruit fly (*Drosophila melanogaster*—Dm), and human (*Homo sapiens*—Hs). The clades of E2s related to yeast Ubc6 and Ubc7 (and the *Arabidopsis* E2 subfamilies) are indicated.

**Figure 7 plants-09-00723-f007:**
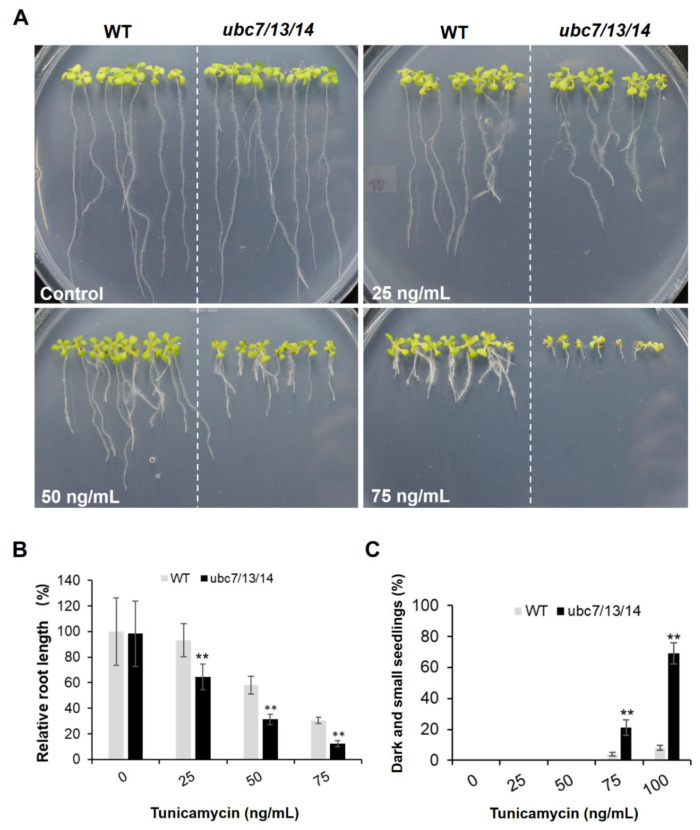
Sensitivity of the *ubc7/13/14* seedlings to tunicamycin. (**A**,**B**) The WT and *ubc7/13/14* mutant seeds were sterilized, stored at 4 °C, and then plated onto the plates with different concentrations of tunicamycin. In each plate, seven WT and seven mutant seeds were planted side by side. There were four replicate plates for each treatment and they were placed vertically in a tissue culture chamber. (**A**) Representative images of WT and mutant seedlings (14-day-old) on control, 25, 50 and 75 ng/mL tunicamycin plates. (**B**) Relative root length. Images were taken on the 14th day after seed plating and the average root length of the seven seedlings from each plate obtained (root lengths measured using ImageJ). The root length of WT seedlings in the normal plates was set to 100%. (**C**) Survey of severely inhibited seedlings (dark brown and small) was taken on the 14th day after seed plating. For each concentration, each plant line had four replicate plates with each plate having 50 seeds. Plates were placed horizontally in a tissue culture chamber. The significance of difference between the mutant and WT was analyzed using the Student’s *t*-test (error bar = SE; **: *p* < 0.01).

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
