# Peer review of "Arabidopsis Ubiquitin-Conjugating Enzymes UBC7, UBC13, and UBC14 Are Required in Plant Responses to Multiple Stress Conditions"

_plants, 2020, doi:10.3390/plants9060723_

Round 1

Reviewer 1 Report

In this work, the authors isolate and characterize loss-of-function mutants in 3 related Arabidopsis ubiquitin conjugating enzymes. They show that loss of function mutants in these individually cause modest stress response hypersensitivity and when all three are eliminated together, this effect is stronger. 

The data are clear and convincing. This work is useful to the community. [Better yet would be a statement that the singles, double, and triple mutants seeds are deposited in a seed bank so that others could use them (and include the name of the seed bank)!]

 Here are a few concerns that should be addressed in my opinion.

  1. The authors argue that these three UBCs are not functionally redundant because each single mutant has a stress-induced phenotype, stating that “if their functions are completely redundant, the single ubc7, ubc13, ubc14 mutants would not be expected to to show clear phenotypes as observed.” While this is a reasonable conclusion, could the authors consider the possibility that there is a quantitative aspect to the stress response? Perhaps missing one of these UBCs reduces the overall level of these enzymes such that a stress response is impaired with simply the reduction in the number? I don’t think this argument can be eliminated completely.  I don’t think the argument that they are biochemically and functionally redundant could be eliminated.
  2. In the discussion, I found the initial description of the yeast comparisons confusing. The authors write that plant UBC32 is “similar to yeast Ubc7 in the ERAD pathway [15}.”  Then they say that UBC7, 13,14 have similarity with yeast Ubc7 and therefore are likely to share functionality in the ERAD pathway.  This is confusing…  In the tree, Figure 6, UBC32 is in the same branch as yeast Ubc6, not 7.  Then later a more complete explanation is given. I suggest that the authors remove the direct reference to similarity (I am assuming this is sequence similarity?) between UBC32 (Arabidopsis) and yeast Ubc7. The subsequent explanation clarifies much. If a previous study was mistaken about identity perhaps that could be mentioned?  Could they modify the discussion to be a bit clearer and less redundant?

Minor

  1. Figure 1A. Please add a scale bar.
  2. Figure 1A Are the 5’ and 3’ UTRs not known for UBC7?
  3. Figure 1B,C. Bp indicating the sizes of a few of the markers next to them on the gel would be appreciated.
  4. Line 100. The text refers to PCR with T-DNA band and a flanking band and cites Figure 1B. That data are not shown in Figure 1B. Only the gene primer reactions are shown. These should be shown- immediately below, switching the RT-PCR data to 1D?
  5. Line 101, 104 the term perspective should be respective.
  6. Figure 4,5,7. I suggest that the axis be labeled as % bleached. Does one know that they are dead?  White seedlings have been known to grow.
  7. Figure 7C. There is an errant ** in the y axis label.  
  8. Figure S4 needs error bars. this looks like idealized data rather than real data.
  9. Figure 6. The names of the yeast proteins are in all caps, should they be written by the yeast nomenclature rules?

Reviewer 2 Report

General comments

The authors characterized ubc7, ubc13 and ubc14 T-DNA single mutants and generated the 3 derived combinations of double mutants as well as the triple mutant ubc7/13/14. The authors examined the impact of the triple mutation on the response to various hormones and abiotic stresses. This study highlights the functional importance of these three E2 proteins in specific stress conditions.

The manuscript is generally written in a clear manner, although I think that the following points should be improved.

Comment for the introduction

Lines 77-80: To avoid any potential confusion, please precise here that UBC13, UBC13A and UBC13B correspond to 3 independent genes.

Comments for the results

Line 89: “singe” has to be replaced by “single”
Line 96: “UB14” has to be replaced by “UBC14”
Lines 101 and 104: “respective” instead of “perspective”?

Comments about the figures

Figure 1A:
- Please precise the ATG position on the schematics.
- Please indicate the length (kb) of each genomic structure.

Figure 1B-C:
- Please indicate the size of the bands obtained, or specify the ladder used.

Figure S4:
- In the figure legend, it is mentioned that “the relative root length data from Figure
4 are presented”, but it is actually from the Figure 3 I understand.
- Overall, I think that the Figure S4 is not required as it presents exactly the same
quantitative data than the Figure 3, and the Figure 3 presents statistics that already
allow the reader to easily appreciate the difference between wild-type and mutant.
Therefore, I suggest to remove the Figure S4.

Figure 3:
- “Salinity” should be replaced by “NaCl”

Comments for the discussion

Line 185: “UBC7, UBC13 and UBC14 (three members of subfamily XIV)” is in
contradiction with the statement “Arabidopsis UBC7, UBC13 and UBC14 are three
members in the subfamily V” in the introduction line 80. Please correct as needed.

Comments for the materials and methods

Line 262: Please precise if Col-0 is the ecotype background of the mutant plants.
Line 277: Please indicate if a DNase treatment was performed before cDNA synthesis.
Lines 279-280: Please complete the sentence “For RT-PCR.”
Line 322: Please precise the method of transformation.
